# Learning Easily Updated General Purpose Text Representations with Adaptable Task-Specific Prefixes

**Kuan-Hao Huang**[*†]  **Liang Tan**[‡]  **Rui Hou**[‡]
**Sinong Wang**[‡]  **Amjad Almahairi**[‡]  **Ruty Rinott**[‡]
[†]University of California, Los Angeles  [‡]Meta AI
khhuang@cs.ucla.edu
{liangtan, rayhou, sinongwang, aalmah, ruty}@meta.com

## Abstract

Many real-world applications require making multiple predictions from the same text. Fine-tuning a large pre-trained language model for each downstream task causes computational burdens in the inference time due to several times of forward passes. To amortize the computational cost, *freezing* the language model and building lightweight models for downstream tasks based on *fixed* text representations are common solutions. Accordingly, how to learn fixed but general text representations that can generalize well to unseen downstream tasks becomes a challenge. Previous works have shown that the generalizability of representations can be improved by fine-tuning the pre-trained language model with some source tasks in a multi-tasking way. In this work, we propose a prefix-based method to learn the fixed text representations with source tasks. We learn a task-specific prefix for each source task independently and combine them to get the final representations. Our experimental results show that prefix-based training performs better than multi-tasking training and can update the text representations at a smaller computational cost than multi-tasking training.

## 1 Introduction

Fine-tuning large pre-trained language models for downstream tasks has become a popular solution in natural language processing (Devlin et al., 2019; Liu et al., 2019b). Although effective, fine-tuning the *whole* language model might cause some computational burdens, especially for those applications that require making multiple predictions from the same text. For example, given a Facebook post, we may want to know its topic, predict its sentiment, extract events, and decide if it contains offensive words, etc. If we train a separate model for each task, we may have a latency during inference time due to several times of forward passes,

which causes computational issues especially when the number of downstream tasks grows.

To amortize the computational cost, several works (Peters et al., 2019; Du et al., 2020) consider *freezing* the language model as the text encoder. They use the frozen language model to obtain the *fixed* representations for a text and build a lightweight model for each downstream task on top of such *fixed* representations. They show that by fine-tuning a pre-trained language model with some *source tasks* in a multi-tasking way, the generated fixed representations capture general information and generalize well for unseen *target tasks*.

In this work, we consider the same goal and make the inference computationally efficient. We aim to learn *fixed* representations from some source tasks that can generalize to unseen target tasks. Instead of multi-tasking training, we propose a new method based on the prefix tuning (Li and Liang, 2021; Liu et al., 2022b) to learn the fixed representations. Specifically, we learn a task-specific prefix for each source task independently. During inference time, all the task-specific prefixes are combined together to produce the final fixed representations. Since those prefixes carry task-specific information, the generated fixed representations capture enough information to generalize to unseen target tasks.

Compared to multi-tasking training, the advantage of prefix-based training is that the fixed text representations can be easily updated at a small computational cost. For example, if we want to add source tasks, we can simply train new prefixes for those tasks without re-training the whole model. Similarly, if we want to remove source tasks, we can directly disable the corresponding prefixes during inference time. In contrast, multi-tasking training requires re-training the whole model, which is less flexible and computationally expensive.

Our experimental results show that prefix-based training performs better than multi-tasking train-

---

[*]Work done while interning at Meta AI.

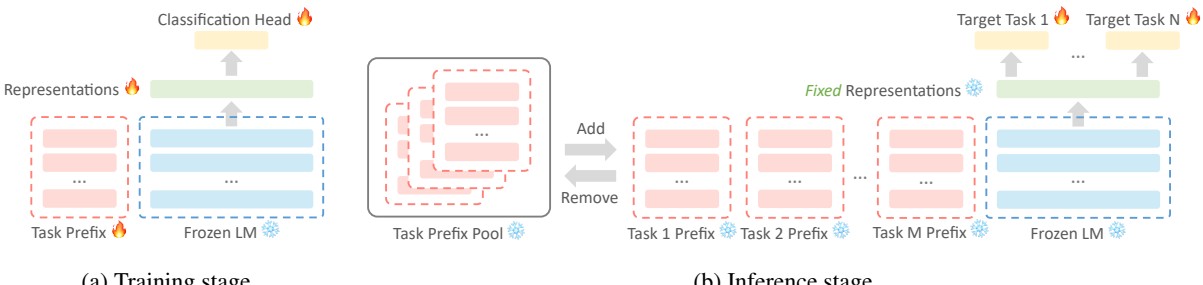

| (a) Training stage. | (b) Inference stage. |

Figure 1: Overview of prefix-based training. (a) We train a task-specific prefix for each source task. (b) All task-specific prefixes are combined together to obtain the fixed text representations. The text representations can be easily updated by adding or removing task-specific prefixes. The *snowflake* symbol and the *fire* symbol indicate whether the module is frozen or not.

ing in terms of transferring knowledge from source tasks to target tasks. In addition, we design two experiments to highlight the flexibility of prefix-based training to easily update the text representations. All the results suggest that prefix-based training can be a promising approach to this research direction.

## 2 Related Work

**General purpose text representations.** Large pre-trained language models are usually used for extracting text representations (Devlin et al., 2019; Liu et al., 2019b; Lewis et al., 2020). Several approaches consider self-supervised learning to improve the quality of representations (Yan et al., 2021; Liu et al., 2021a; Gao et al., 2021; Chuang et al., 2022). To improve the generalization for unseen tasks, some works consider additional source tasks to learn the representations, such as natural language inference corpus (Conneau et al., 2017; Cer et al., 2018; Reimers and Gurevych, 2019) and paraphrase pairs (Huang et al., 2021). Recently, frozen text representations have caught attention for amortizing the computational cost in real-world applications (Peters et al., 2019; Du et al., 2020).

**Prefix tuning and prompt tuning.** Recently, prefix tuning and prompt tuning become popular ways to learn parameter-efficient models. Early works usually consider discrete prompts, where the prompts consist of real words (Shin et al., 2020; Schick and Schütze, 2021a,b; Scao and Rush, 2021). Later works study soft prompts, where the prompt words are learnable (Li and Liang, 2021; Liu et al., 2021b; Lester et al., 2021a; Qin and Eisner, 2021; Liu et al., 2022b). Several studies have shown that prefixes and prompts can effectively capture the key information about the tasks (Liu et al., 2022a; Hsu et al., 2023; Wan et al., 2023;

Cao et al., 2023). Our work is motivated by the fact that those learnable parameters can be viewed as embeddings for transferring knowledge or representing tasks (Vu et al., 2022; Asai et al., 2022; Zhou et al., 2022).

## 3 Method

### 3.1 Problem Setup

Our goal is to learn *fixed* text representations that perform well on unseen target tasks. During the training stage, we consider $M$ source tasks $T_1^s, T_2^s, ..., T_M^s$ to learn text representations. In the inference stage, we train a *lightweight* classifier for each target task $T_1^t, T_2^t, ..., T_N^t$ based on the learned *fixed* text representations. That is, only the lightweight classifier will be trained while the text representations remain the same to reduce the computational burden during the inference time. Additionally, we expect the learned representations can be easily updated (e.g., add/remove/update source tasks) at a small computational cost.

### 3.2 Training Stage

As illustrated by Figure 1a, we learn a task-specific prefix (Li and Liang, 2021; Liu et al., 2022b) for each source task. It is worth noticing that every task-specific prefix is learned *independently*. Specifically, we follow the implementation of P-Tuning v2 (Liu et al., 2022b) and consider the soft prompt tuning (Lester et al., 2021b) for each Transformer layer. For each layer, we learn an additional key matrix $K_p = \{\mathbf{k}_1, ..., \mathbf{k}_l\}$ and an additional value matrix $V_p = \{\mathbf{v}_1, ..., \mathbf{v}_l\}$, where $l$ is the length of the prefix. When computing the attentions for each layer, we concatenate the additionally learned key matrix $K_p$ and value matrix $V_p$ with the original key matrix $K$ and value matrix $V$. That is, we use $K' = K_p \oplus K$ and $V' = V_p \oplus V$

to calculate the scaled dot-product attention. More training details can be found in Appendix A.1.

The final task-specific prefix $P$ consists of those learned key matrices $\{K_p^{(1)}, ..., K_p^{(L)}\}$ and value matrices $\{V_p^{(1)}, ..., V_p^{(L)}\}$ for all layers, where $L$ is the number of layers. Since the language model is frozen during training, we expect that all task-specific information is captured by the prefix $P$.

## 3.3 Inference Stage

Assuming the task-specific prefixes we learn for the source tasks $T_1^s, T_2^s, ..., T_M^s$ are $P_1, P_2, ..., P_M$, we concatenate them to be a large prefix $P^*$. In other words, for each Transformers layer, we use $K^* = K_1' \oplus K_2' \oplus ... \oplus K_M'$ and $V^* = V_1' \oplus V_2' \oplus ... \oplus V_M'$ to calculate the attention and compute the final text representations. We then train classifiers for target tasks on top of the same *fixed* text representations, as illustrated by Figure 1b. Appendix A lists more details.

As mentioned in Section 1, to reduce the computational cost, all the prefixes and the language model are frozen during the inference stage. However, the task-specific prefixes can still pass the task-specific information via the learned key matrices and value matrices when calculating attention. Therefore, the final text representations contain necessary information about source tasks that can be transferred to unseen target tasks.

## 3.4 Comparison to Multi-Tasking Training

Multi-tasking learning (Collobert and Weston, 2008; Zhang and Yang, 2017; Ahmad et al., 2018; Liu et al., 2019a; Zhou et al., 2021) is a common way to incorporate the knowledge of several source tasks into language models. It fine-tunes a language model with multiple objectives for source tasks at the same time. Compared to multi-tasking training, our prefix-based training has several advantages for obtaining fixed text representations.

The biggest advantage of prefix-based training is that the text representations can be easily updated at a small computational cost. For example, if we want to add (update) some source tasks, we can simply train new (update) prefixes for those tasks. Similarly, if we would like to remove some source tasks, we can directly disable the corresponding prefixes during the inference time. Compared to multi-tasking training, which needs to re-train the whole language model when adding/updating/removing source tasks, prefix-based training has great flexi-

| Dataset | Task Type | # of Train |
|---|---|---|
| *Source Tasks* | | |
| MNLI | Natural Language Inference (NLI) | 393K |
| QNLI | Natural Language Inference (NLI) | 105K |
| QQP | Paraphrase Identification (PI) | 364K |
| SST-2 | Sentiment Analysis (SA) | 66K |
| Yelp-2 | Sentiment Analysis (SA) | 540K |
| ReCoRD | Question Answering (QA) | 101K |
| WinoGrande | Commonsense Reasoning (CR) | 40K |
| *Target Tasks* | | |
| RTE | Natural Language Inference (NLI) | 2.5K |
| MRPC | Paraphrase Identification (PI) | 3.7K |
| CR | Sentiment Analysis (SA) | 2.3K |
| MR | Sentiment Analysis (SA) | 6.4K |
| MPQA | Sentiment Analysis (SA) | 6.4K |
| BoolQ | Question Answering (QA) | 9.4K |
| MultiRC | Question Answering (QA) | 27K |
| CosmosQA | Commonsense Reasoning (CR) | 25K |

Table 1: Datasets.

bility for updating text representations.

In addition, prefix-based is faster and easier. Since all task-specific prefixes are trained independently they can be trained in parallel, using different parameters (e.g. learning rates). This solves the difficulty of multi-task training where it is hard to find a good configuration for all source tasks, as different tasks have different properties, e.g. tasks have varying sizes of training examples, which causes some tasks to be dominated by others (Liang and Zhang, 2020; Mao et al., 2022).

## 4 Experiments

We conduct experiments to show the potential of prefix-based training and its flexibility for updating text representations.

### 4.1 Tasks

We consider different types of NLP tasks, including natural language inference, paraphrase identification, sentiment analysis, question answering, and commonsense reasoning, as listed in Table 1. The source tasks include the following 7 datasets with more than 40K annotations: MNLI (Williams et al., 2018), QNLI (Demszky et al., 2018), QQP (Wang et al., 2019), SST-2 (Socher et al., 2013), Yelp-2 (Charwad et al., 2015), ReCoRD (Zhang et al., 2018), and WinoGrande (Sakaguchi et al., 2020). The target tasks include the following 8 relatively small datasets: RTE (Giampiccolo et al., 2007), MRPC (Dolan et al., 2004), CR (Hu and Liu, 2004), MR (Pang and Lee, 2005), MPQA (Wiebe et al., 2005), BoolQ (Clark et al., 2019), MultiRC (Khashabi et al., 2018), and CosmosQA (Huang

| Method | Freeze | RTE | MRPC | CR | MR | MPQA | BoolQ | MultiRC | CosmosQA | Avg. |
|---|---|---|---|---|---|---|---|---|---|---|
| *Upper bound (computationally expensive)* | | | | | | | | | | |
| Fine-tuning the whole language model | ✗ | 86.38 | 88.01 | 93.59 | 91.19 | 91.93 | 85.28 | 80.80 | 77.77 | 86.87 |
| *Lower bound, without source task pre-training* | | | | | | | | | | |
| Fine-tuning with frozen language model | ✓ | 58.59 | 77.28 | 91.66 | 87.99 | 90.29 | 70.53 | 68.74 | 57.27 | 75.29 |
| *With source task pre-training* | | | | | | | | | | |
| Multi-tasking training | ✓ | 82.16 | 84.90 | 91.03 | **90.60** | 90.83 | 79.44 | 74.00 | 66.53 | 82.44 |
| Prefix-based training (ours) | ✓ | **84.86** | **85.37** | **92.03** | 90.08 | **91.05** | **80.84** | **75.66** | **70.90** | **83.85** |

Table 2: 5-run average results for transfer setting. Prefix-based training performs better than multi-tasking training.

| Method | Freeze | RTE | MRPC | CR | MR | MPQA | BoolQ | MultiRC | CosmosQA | Avg. |
|---|---|---|---|---|---|---|---|---|---|---|
| Multi-tasking training | ✓ | 82.16 | 84.90 | 91.03 | 90.60 | 90.83 | 79.44 | 74.00 | 66.53 | 82.44 |
| Prefix-based training | ✓ | 84.86 | 85.37 | 92.03 | 90.08 | 91.05 | 80.84 | 75.66 | 70.90 | 83.85 |
| - Removing at most 1 task | ✓ | 86.76 | 86.21 | 92.05 | 90.64 | 91.28 | 81.65 | **76.12** | 71.11 | 84.48 |
| - Removing at most 2 tasks | ✓ | **87.30** | **86.67** | **93.05** | **90.76** | **91.70** | **82.18** | **76.12** | **72.00** | **84.97** |

Table 3: Best results from all combinations of removal. Removing hurtful source tasks leads to improvements.

et al., 2019).

For those datasets with the standard train, dev, and test split, we follow the standard split for training. For those datasets without the standard split (e.g., GLUE tasks), we randomly split 1/3 examples from the dev set as the internal dev set and use the rest 2/3 examples as the testing set.

## 4.2 Baselines for Comparison

We compare our proposed prefix-based training with multi-tasking training (Collobert and Weston, 2008; Zhang and Yang, 2017; Ahmad et al., 2018; Liu et al., 2019a; Zhou et al., 2021). Both approaches are trained on the same source tasks and use pre-trained RoBERTa-large (Liu et al., 2019b). Then, we *freeze* both models and get *fixed* text representations as the features to train classifiers for the target tasks. Please refer to Appendix A for more details. To analyze the influence of source tasks and fixed representations, we additionally consider two simple fine-tuning baselines: fine-tuning the whole language model and fine-tuning with the frozen language model. Note that they directly use RoBERTa-large without training on source tasks.

## 4.3 Results for Transfer Setting

From Table 2, we first observe that simple fine-tuning without freezing text representations performs much better than with freezing. This indicates that although fixed representations reduce the computational cost, they largely limit the power of pre-trained language models. However, if we freeze the representations, with training on source tasks, we see an overall improvement for all target tasks, which shows the importance of knowledge transfer from source tasks to target tasks.

Prefix-based training consistently performs better than multi-tasking training, especially for those target tasks that require high-level understanding, such as natural language inference and commonsense reasoning. We hypothesize that those types of tasks are more difficult than others and might be dominated by other simpler tasks, such as sentiment analysis, during multi-tasking training. Therefore, multi-tasking training cannot transfer knowledge from those types of tasks well. In contrast, prefix-based training has promising performance for all types of tasks.

## 4.4 Flexibility for Updating Representations

As mentioned in Section 3, one advantage of prefix-based training is that the text representations can be easily updated with a small computational cost. We conduct two experiments to verify this merit.

**Removing hurtful source tasks.** Prefix-based training gives us an easy way to disable some source tasks during the inference stage — just removing the corresponding prefixes without retraining. Therefore, we can easily find out some hurtful source tasks for a particular target task by removing different combinations of source tasks.

Table 3 shows the best results from all combinations of removal. We observe improvements when removing hurtful source tasks. For example, removing SST-2 task is helpful for natural language

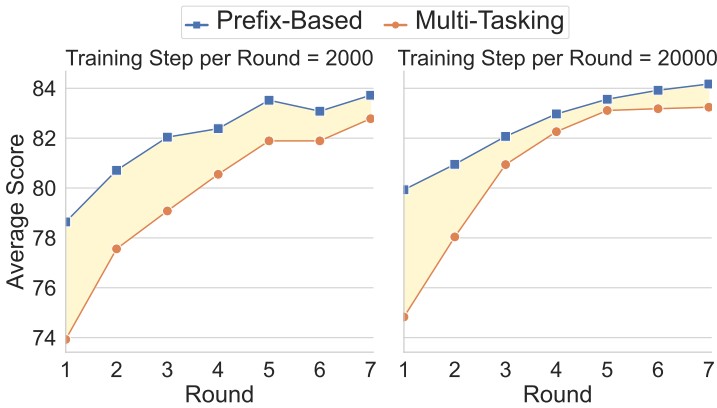

Figure 2: Results of sequentially adding source tasks.

inference and commonsense reasoning tasks, while removing ReCoRD task is helpful for sentiment analysis tasks. This experiment shows that we can use the flexibility of prefix-based training to find out hurtful source tasks and improve performance.

**Adding new source tasks.** To mimic the situation when new source tasks are introduced, instead of training on all source tasks together, we sequentially add one source task every round and update both models of prefix-based training and multi-tasking training. For prefix-based training, we train a prefix for the new task. For multi-tasking training, we re-train the whole model with all existing source tasks. To fairly compare the two approaches, we fix the number of training steps per round. We run 8 repeats of experiments with different orders to add tasks and report the average performance on target tasks for each round in Figure 2. Experimental details can be found in Appendix B.1.

Overall, prefix-based training is better than multi-tasking training for every round. When the number of training steps becomes smaller, prefix-based training still has similar performance while the performance of multi-tasking training drops a lot. This is because prefix-based training can use all training steps for the new task while multi-tasking training has to share the training steps over all existing source tasks. This experiment shows the flexibility of prefix-based training for updating text representations at a smaller computational cost.

## 5 Conclusion

We focus on learning *fixed* text representations with source tasks that can generalize well to unseen target tasks. We propose a prefix-based training method that learns a task-specific prefix for each source task. The fixed text representations are com-

puted by combining all task-specific prefixes together. Our experimental results show that prefix-based training performs better than multi-tasking training and can update representations at a smaller computational cost than multi-tasking training.

## Limitations

In this work, our goal is to prove the concept that prefix-tuning training is better than multi-tasking training in terms of transferring knowledge and updating fixed text representations. We try to include as many tasks as possible in our experiments. However, we understand that there might be some differences between our experimental settings and real-world cases. For example, the current experiments are limited to text-understanding tasks. Some other types of tasks, such as structure predictions and syntax-related tasks, are not considered in the current version. Also, in the real-world case, the number of source tasks and target tasks can be larger. In this work, we provide a proof of concept and demonstrate the potential benefits of the prefix-based training method. Increasing the number of tasks is therefore considered as our future study.

## Broader Impacts

Our model is based on large pre-trained language models. It is known that the models trained with a large text corpus may capture the bias reflecting the training data. Therefore, it is possible that the predictions produced by our model inherit the bias of pre-trained language models. We suggest to carefully examining the potential bias before applying our method to any real-world applications.

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

## A  Training Details

All the models are trained with NVIDIA Tesla V100 GPU.

### A.1  Details for Training Task-Specific Prefix

We follow the implementation of P-tuning v2 (Liu et al., 2022b) and set the prompt length to 5. The Batch size is 16 for all source tasks and the number of epoch is 40. We use Adam optimizer with the learning rate being 5e-3 and the weight decay being 1e-5. The classification head is one layer of MLP.

We notice that in the original implementation of P-Tuning v2, they add positional encoding to prefix. However, as we will concatenate several task-specific prefix together during the inference time, we remove the positional encoding when training the prefix to avoiding position mismatch between training and inference.

### A.2  Details for Training Target Tasks

Following previous work (Du et al., 2020), we train one attention layer on top of the fixed text representations and train one layer of MLP based on the CLS token representation for each target task. The batch size is 32 and the number of epoch is 80. We use Adam optimizer with the learning rate being 1e-4 and the weight decay being 1e-5.

### A.3  Details for multi-tasking

We uniformly sample data from source tasks for every batch. The batch size is 16 and the number of training steps is 400000. We use Adam optimizer with the learning rate being 1e-5 and the weight decay being 1e-5. The classification head is one layer of MLP.

## B  Experimental Details

### B.1  Sequence Order

We consider the following 8 sequence to add the source tasks sequetially.

- MNLI, QQP, ReCoRD, QNLI, SST-2, WinoGrande, Yelp-2

- Yelp-2, MNLI, ReCoRD, SST-2, QNLI, WinoGrande, QQP

- QQP, ReCoRD, MNLI, SST-2, QNLI, WinoGrande, Yelp-2

- ReCoRD, Yelp-2, WinoGrande, QQP, MNLI, SST-2, QNLI

- SST-2, QNLI, ReCoRD, MNLI, WinoGrande, Yelp-2, QQP

- QNLI, WinoGrande, MNLI, QQP, SST-2, ReCoRD, Yelp-2

- WinoGrande, SST-2, Yelp-2, QQP, QNLI, ReCoRD, MNLI

- QNLI, Yelp-2, QQP, ReCoRD, WinoGrande, MNLI, SST-2