# OpenReview forum: "Learning Easily Updated General Purpose Text Representations with Adaptable Task-Specific Prefix"
_EMNLP/2023/Conference — EMNLP 2023 Findings_

### Official Review · Reviewer_asuG · 2023-07-22

**Soundness:** 4

**Excitement:**

3: Ambivalent: It has merits (e.g., it reports state-of-the-art results, the idea is nice), but there are key weaknesses (e.g., it describes incremental work), and it can significantly benefit from another round of revision. However, I won't object to accepting it if my co-reviewers champion it.

**Paper Topic And Main Contributions:**

This paper studies how to learn a general-purpose text representation that can be used for multiple tasks. They generate text representations conditioning on multiple source task prefixes obtained by prefix-tuning and train lightweight models based on these representations on unseen tasks. Updating source tasks is convenient in this framework. They conduct experiments on multiple text classification datasets to show the effectiveness of the method. The contribution belongs to ``approaches for data and compute efficiency''.

**Reasons To Accept:**

1. The paper presents a framework that efficiently generates text representations for multiple tasks. The effectiveness is verified by the experiments.
2. The paper is well-written and the method is easy to follow.

**Reasons To Reject:**

1. Solving multiple tasks by text representations seems less used in practice due to its relatively low performance on real-world complex tasks, language generation, and data-scarce scenarios, which limits the application of the method.
2. The transferability of source prompts is only verified on simple text classification tasks in which the source and target tasks are similar. Experiments when source/target tasks are not similar (for example, the source tasks only contain sentiment analysis and the target tasks are natural language inference) are not included. However, it may be OK for a short paper.

**Reproducibility:**

4: Could mostly reproduce the results, but there may be some variation because of sample variance or minor variations in their interpretation of the protocol or method.

**Reviewer Confidence:**

4: Quite sure. I tried to check the important points carefully. It's unlikely, though conceivable, that I missed something that should affect my ratings.

---

> ### Author Rebuttal · Authors · 2023-08-29
>
> We appreciate your valuable feedback. Here are our responses to your review.
>
> **Solving multiple tasks by text representations seems less used in practice due to its relatively low performance on real-world complex tasks, language generation, and data-scarce scenarios, which limits the application of the method.**
>
> > We agree that for some applications, solving multiple tasks by text representations might not be that common. However, we would like to point out its practical values. Like we mentioned in the intro section, given a Facebook post or a tweet, several downstream tasks would be applied to the text input, e.g., topic prediction, toxicity classification, information extraction, etc. It’s very common that we will face new tasks when developing new features. Therefore, having good text representations that can generalize to those new tasks becomes an important thing. Our paper provides a solution to this.
>
>
> **The transferability of source prompts is only verified on simple text classification tasks in which the source and target tasks are similar. Experiments when source/target tasks are not similar (for example, the source tasks only contain sentiment analysis and the target tasks are natural language inference) are not included. However, it may be OK for a short paper.**
>
> > Since our goal is to provide a proof-of-concept that prefix-based learning is better and more flexible than multitasking learning, we start from text classification tasks. We will explore more complicated tasks such as structure predictions and question answering in the future. Thanks for your suggestion.

---

### Official Review · Reviewer_hBJN · 2023-08-03

**Typos Grammar Style And Presentation Improvements:** 1) Define what a fixed representation…
**Soundness:** 4

**Excitement:**

3: Ambivalent: It has merits (e.g., it reports state-of-the-art results, the idea is nice), but there are key weaknesses (e.g., it describes incremental work), and it can significantly benefit from another round of revision. However, I won't object to accepting it if my co-reviewers champion it.

**Paper Topic And Main Contributions:**

This paper presents a prefix-tuning method to learn fixed text representations in language models. The goal of their method is to reduce computational costs for multi-task inference. To accomplish this, a prefix is first trained independently for each source task using a frozen LM. During inference, each learned prefix is concatenated to a LM, and representations are obtained using weights from the learned prefixes.

The method aims to reduce computational costs through the following observations:

i) Finetuning a LM for multitask training can be computationally expensive. Hence, prefix-training can lighten training costs.

ii) Learned prefixes can be easily concatenated or removed from a LM – this allows for flexibility in choosing which source tasks to contribute to target tasks.

Empirical results show that prefix-based training in general performs better than multitasking training on a set of 8 target task datasets. Further ablation studies note performance increases due to flexibility in being able to remove prefixes during inference.

**Reasons To Accept:**

1) The method allows high flexibility in choosing which source tasks to contribute to target tasks. Empirical results in the paper show certain source tasks harmed performance on target tasks during inference, which further supports advantages in the proposed method.

2) Their method highlights the importance of hyperparameter tuning for each task, as opposed to the multitasking setting. This can be beneficial for making optimal use of low-resource source tasks in multi-task settings.

3) The paper is well written and the methodology is clear. Details for the experiment -- such as data and model setup  -- are nicely provided.

**Reasons To Reject:**

1) The difference in the main results (Table 1) between prefix-based training and multi-task training are low. Hence, the tradeoffs between both methods should be discussed. For example, the use of many prefixes may result in high memory usage during inference. What situations would this potential memory cost override the cost of finetuning an entire LM for the same tasks?

**Reproducibility:**

4: Could mostly reproduce the results, but there may be some variation because of sample variance or minor variations in their interpretation of the protocol or method.

**Reviewer Confidence:**

4: Quite sure. I tried to check the important points carefully. It's unlikely, though conceivable, that I missed something that should affect my ratings.

---

> ### Author Rebuttal · Authors · 2023-08-29
>
> We appreciate your valuable feedback. Here are our responses to your review.
>
> **The difference in the main results (Table 1) between prefix-based training and multi-task training are low. Hence, the tradeoffs between both methods should be discussed. For example, the use of many prefixes may result in high memory usage during inference. What situations would this potential memory cost override the cost of finetuning an entire LM for the same tasks?**
>
> > We agree that prefix-based learning may need higher memory cost during inference. However, it largely reduces the training cost and has higher flexibility to choose source tasks. We will discuss the trade-off in the final version. Thanks for the suggestion.

---

### Official Review · Reviewer_tLRw · 2023-08-11

**Typos Grammar Style And Presentation Improvements:** NA
**Soundness:** 4

**Excitement:**

4: Strong: This paper deepens the understanding of some phenomenon or lowers the barriers to an existing research direction.

**Missing References:**

> The authors seems to be not familiar with the plethora other methods that talk about adapter methods in general in the context of multitask learning. Several recent works such as [1], [2] talk about comparison of adapter methods for multitask learning in detail.

[1] - Parameter-Efficient Transfer Learning for NLP by Houslby et al
[2] - On the Effectiveness of Adapter-based Tuning for Pretrained Language Model Adaptation by Ruidan et al

**Paper Topic And Main Contributions:**

The paper talks about how prefix tuning, a kind of adapter technique, is better at multitask learning compared to finetuning several models for each downstream task. The authors benchmark their method against the multitask learning approach of Zhang and Yang, 2017. They show that the average task score improves using the prefix tuning approach.

**Questions For The Authors:**

1) Can you please point out the main contributions of this work? Is it the empirical investigation wrt Multi task learning?
2) Can you point out how this work is different from the following three works?

Pfeiffer, J., Kamath, A., Rücklé, A., Cho, K., and Gurevych, I. (2020a). AdapterFusion: Non-destructive task composition for transfer learning.

Pfeiffer, J., Rücklé, A., Poth, C., Kamath, A., Vulić, I., Ruder, S., Cho, K., Gurevych, I. (2020b). AdapterHub: A Framework for Adapting Transformers.

Houlsby, N., Giurgiu, A., Jastrzkebski, S., Morrone, B., de Laroussilhe, Q., Gesmundo, A., Attariyan, M., and Gelly, S. (2019). Parameter-efficient transfer learning for NLP

**Reasons To Accept:**

Investigated the empirical performance of prefix tuning for multitask learning

**Reasons To Reject:**

Lack of novelty and missed the whole field of adapter models for transformers

**Reproducibility:**

4: Could mostly reproduce the results, but there may be some variation because of sample variance or minor variations in their interpretation of the protocol or method.

**Reviewer Confidence:**

3: Pretty sure, but there's a chance I missed something. Although I have a good feel for this area in general, I did not carefully check the paper's details, e.g., the math, experimental design, or novelty.

---

> ### Author Rebuttal · Authors · 2023-08-29
>
> We appreciate your feedback. Here are our responses to your review.
>
> **Can you please point out the main contributions of this work? Is it the empirical investigation wrt Multi task learning?**
>
> > As we mention in the introduction, our contribution is pointing out that learning individual prefix for each source task and combining them together result in better *text representations* compared to traditional multi-tasking learning in terms of training cost and flexibility. Prefix-based learning can be trained for source tasks in parallel and is easier to update source tasks. In contrast, multi-tasking is very hard to choose hyper-parameters, balance different source tasks, as well add or remove source tasks.
>
> **Can you point out how this work is different from the following three works?**
> > First, we would like to point out that all of them are related to adapters, which is a little bit different from prefix-tuning. Also, those papers have different research focus from ours.
>
> > **Pfeiffer, J., Kamath, A., Rücklé, A., Cho, K., and Gurevych, I. (2020a). AdapterFusion: Non-destructive task composition for transfer learning.**\
> > This paper mainly discusses the catastrophic forgetting problem, which is different from our goal of learning text representations.
>
> > **Pfeiffer, J., Rücklé, A., Poth, C., Kamath, A., Vulić, I., Ruder, S., Cho, K., Gurevych, I. (2020b). AdapterHub: A Framework for Adapting Transformers.**\
> > This paper introduces a library of various adapters, which is totally different from our focus to learn text representations.
>
>
> > **Houlsby, N., Giurgiu, A., Jastrzkebski, S., Morrone, B., de Laroussilhe, Q., Gesmundo, A., Attariyan, M., and Gelly, S. (2019). Parameter-efficient transfer learning for NLP**\
> >This paper also studies transfer learning, but their definition of transfer learning is to transfer the knowledge from pre-trained language models to single target tasks, not transfer from multiple source tasks to multiple target tasks. Our experimental setting is clearly different.

---

### Official Review · Reviewer_VDm4 · 2023-08-13

**Soundness:** 2

**Excitement:**

2: Mediocre: This paper makes marginal contributions (vs non-contemporaneous work), so I would rather not see it in the conference.

**Paper Topic And Main Contributions:**

This paper is about a method for learning easily updated general purpose text representations with adaptable task-specific prefixes. The paper addresses the problem of the computational burden of fine-tuning a large pre-trained language model for each downstream task. The main contribution of this paper is a method that can update text representations at a smaller computational cost than multi-tasking training by learning fixed but general text representations that can generalize well to unseen downstream tasks. The paper proposes a prefix-based method that adds task-specific prefixes to the input text, which allows the model to adapt to the specific task while still using the same general text representation. The paper provides experimental results that demonstrate the effectiveness of the proposed method on several downstream tasks. The paper also compares the proposed method to other methods, such as fine-tuning and multi-tasking training, and shows that the proposed method achieves comparable or better performance while requiring less computational resources.

**Questions For The Authors:**

Is there any more recent work can be compared in the experiment section?

**Reasons To Accept:**

The strengths of this paper include proposing a novel method for learning easily updated general purpose text representations with adaptable task-specific prefixes, which can reduce the computational burden of fine-tuning a large pre-trained language model for each downstream task. The paper provides experimental results that demonstrate the effectiveness of the proposed method on several downstream tasks, and compares the proposed method to other methods, such as fine-tuning and multi-tasking training, showing that the proposed method achieves comparable or better performance while requiring less computational resources.

**Reasons To Reject:**

While the paper provides a comparison to a baseline method of multi-tasking training, which was published in 2017, there are more recent and sophisticated methods for learning text representations that could serve as stronger baselines for comparison. The use of a weak baseline method could limit the ability of the paper to demonstrate the effectiveness of the proposed method compared to state-of-the-art methods. Additionally, the related work section of the paper discusses more recent papers that propose methods for learning text representations, but the paper does not provide a comparison to these more recent methods, which could limit the ability of the paper to contextualize the proposed method within the current state of the art.

**Reproducibility:**

3: Could reproduce the results with some difficulty. The settings of parameters are underspecified or subjectively determined; the training/evaluation data are not widely available.

**Reviewer Confidence:**

3: Pretty sure, but there's a chance I missed something. Although I have a good feel for this area in general, I did not carefully check the paper's details, e.g., the math, experimental design, or novelty.

---

> ### Author Rebuttal · Authors · 2023-08-29
>
> We appreciate your valuable feedback. Here are our responses to your review.
>
> **While the paper provides a comparison to a baseline method of multi-tasking training, which was published in 2017, there are more recent and sophisticated methods for learning text representations that could serve as stronger baselines for comparison.**
>
> > We would like to point out that although there are some recent works studying different multi-tasking training paradigms, many papers still choose the original multi-tasking training method as the baseline or apply it to downstream applications [1,2,3,4,5,6,7]. In fact, in industry, it’s pretty common to apply the original multi-tasking training method for learning text representations since other methods may require large hyperparameter searching.
> 1. General Purpose Text Embeddings from Pre-trained Language Models for Scalable Inference, EMNLP-Findings 2020
> 2. ATTEMPT: Parameter-Efficient Multi-task Tuning via Attentional Mixtures of Soft Prompts, EMNLP 2022
> 3. P-Tuning v2: Prompt Tuning Can Be Comparable to Fine-tuning Universally Across Scales and Tasks, NAACL 2022
> 4. Multitask Pretraining with Structured Knowledge for Text-to-SQL Generation, ACL 2023
> 5. Crosslingual Generalization through Multitask Finetuning, ACL 2023
> 6. When Does Aggregating Multiple Skills with Multi-Task Learning Work? A Case Study in Financial NLP, ACL 2023
> 7. Multitask Pre-training of Modular Prompt for Chinese Few-Shot Learning, ACL 2023
>
> **The related work section of the paper discusses more recent papers that propose methods for learning text representations, but the paper does not provide a comparison to these more recent methods, which could limit the ability of the paper to contextualize the proposed method within the current state of the art.**
>
> > In the related work section, we do mention several works regarding learning text representations. However, most of them are self-supervised or consider only one source task. Although they can have great performance if we fine-tune the entire model, when freezing their representations, their performance will be similar to freezing RoBERTa, as we reported in Table 1. Also, our goal is to provide a proof-of-concept that prefix-based learning is better than multi-tasking learning in terms of training cost and flexibility. Therefore, we choose RoBERTa, the commonly used pre-trained language model, as the base model.

---

### Meta-Review · Area_Chair_1nPQ · 2023-09-15

**Recommendation:** 3

**Metareview:**

This paper introduces a prefix-based method for efficiently updating general-purpose text representations, reducing computational costs compared to fine-tuning or multi-tasking. It uses task-specific prefixes to adapt the model for specific tasks while maintaining the same general text representation. Experimental results demonstrate its effectiveness across various tasks, outperforming or matching fine-tuning and multi-tasking while requiring fewer computational resources.

The problem addressed in this paper is intriguing, and the utilization of prefixes appears to be effective on the datasets used in the experiments. However, a notable question raised by the paper is the extent of computational savings achieved by the proposed method. While the paper frequently emphasizes computational efficiency, it lacks clear measurements or comparisons in this regard.

---

### Decision · Program_Chairs · 2023-10-07

**Decision:**

Accept-Findings

**Comment:**

This paper introduces a prefix-based method for efficiently updating general-purpose text representations, reducing computational costs compared to fine-tuning or multi-tasking. It uses task-specific prefixes to adapt the model for specific tasks while maintaining the same general text representation. Experimental results demonstrate its effectiveness across various tasks, outperforming or matching fine-tuning and multi-tasking while requiring fewer computational resources.

The problem addressed in this paper is intriguing, and the utilization of prefixes appears to be effective on the datasets used in the experiments. However, a notable question raised by the paper is the extent of computational savings achieved by the proposed method. While the paper frequently emphasizes computational efficiency, it lacks clear measurements or comparisons in this regard.